# Prdm16 regulates the postnatal fate of embryonic radial glia via Vcam1-dependent mechanisms

Jiwen Li[1,11], Marlesa I. Godoy [1,11], Yi Lu [1,11], Alice J. Zhang[1], Graciel Diamante[2], Elle Rathbun[3], Min Tian [3], In Sook Ahn[2], Arantxa Cebrian-Silla[4], Arturo Alvarez-Buylla[4], Xia Yang[2,5,6,7], Bennett G. Novitch[5,7,8,9,10], S. Thomas Carmichael [3,5,10] & Ye Zhang [1,5,7,9,10] ✉

The mammalian brain undergoes rapid and extensive neurogenesis during the embryonic stage and limited neurogenesis during the adult stage, which results in ineffective repair of neural circuits in adults. Currently, the molecular mechanisms regulating the postnatal termination of neurogenesis and the disappearance of embryonic radial glia, the neural stem cells (NSCs) responsible for neurogenesis, are largely unknown. Here, we show that genetic deletion of *PR domain-containing 16 (Prdm16)* from NSCs leads to the retention of radial glia in adulthood and prolonged postnatal neuroblast production. Mechanistically, Prdm16 induces a postnatal reduction in Vascular Cell Adhesion Molecule 1 (Vcam1). The extended presence of radial glia and neurogenesis phenotype is rescued in *Prdm16-Vcam1* double knockout mice. These findings demonstrate that the inhibition of Vcam1 by Prdm16 promotes the postnatal cessation of neurogenesis and the disappearance of embryonic radial glia and provide valuable insights for regenerative medicine aimed at treating central nervous system disorders.

Why can't the brains of adult mammals fully regenerate after damage or degeneration? The primary reason lies in their limited capacity for neurogenesis. In adult mice, the production of new neurons is limited, occurring slowly and only in the olfactory bulb and the hippocampal dentate gyrus[1,2]. This is a stark difference from embryonic stages, where rapid neurogenesis occurs throughout all brain regions, resulting in billions of new neurons in a few weeks. The source behind this rapid embryonic neurogenesis is the embryonic neural stem cells (NSCs), also known as radial glia[3–5]. Shortly after birth, however, these cells retract their long radial processes and transform into other cell types[6–10]. Consequently, neurogenesis is terminated throughout the majority of the central nervous system (CNS). For simplicity, we will refer to the postnatal retraction of radial glial processes and transformation of radial glia into adult NSCs and other cell types collectively as the "disappearance" of embryonic radial glia.

How do NSCs "keep time" and stop neurogenesis on schedule? What molecular mechanisms drive the disappearance of radial glia? These questions remain unanswered. While much is known about

[1]Department of Psychiatry and Biobehavioral Sciences, Semel Institute for Neuroscience and Human Behavior, David Geffen School of Medicine at the University of California, Los Angeles (UCLA), Los Angeles, CA, USA. [2]Department of Integrative Biology and Physiology, UCLA, Los Angeles, CA, USA. [3]Department of Neurology, UCLA, Los Angeles, CA, USA. [4]Eli and Edythe Broad Institute for Stem Cell Research and Regeneration Medicine, Department of Neurological Surgery, University of California, San Francisco, San Francisco, CA, USA. [5]Brain Research Institute at UCLA, Los Angeles, CA, USA. [6]Institute for Quantitative and Computational Biosciences at UCLA, Los Angeles, CA, USA. [7]Molecular Biology Institute at UCLA, Los Angeles, CA, USA. [8]Department of Neurobiology, UCLA, Los Angeles, CA, USA. [9]Intellectual and Developmental Disabilities Research Center at UCLA, Los Angeles, CA, USA. [10]Eli and Edythe Broad Center of Regenerative Medicine and Stem Cell Research at UCLA, Los Angeles, CA, USA. [11]These authors contributed equally: Jiwen Li, Marlesa I. Godoy, Yi Lu. ✉e-mail: yezhang@ucla.edu

embryonic and adult neurogenesis, the postnatal transition phase remains underexplored. This gap in understanding postnatal neurogenesis termination hinders the development of stem cell therapies. Current NSC transplantations aimed at treating neuron loss from neurodegeneration, stroke, and brain injury have been unsuccessful in generating enough neurons to significantly improve outcomes. As a result, no FDA-approved NSC therapy exists. Studying how neurogenesis is terminated postnatally could reveal new molecular targets to extend neurogenesis in transplanted NSCs, potentially transforming the treatment landscape of a broad range of neurological disorders.

Radial glia reside in the ventricular-subventricular zone (V-SVZ) on the wall of the lateral ventricle and generate the majority of neurons and glia in the forebrain[1,11–16]. Each NSC/radial glia has a long radial process that frequently extends deep into the adjacent brain parenchyma and guides the migration of new neurons generated in the V-SVZ to their final destination[1,16–25]. From embryonic day 13.5 to 15.5 (E13.5–15.5), a subpopulation of embryonic NSCs/radial glia slows down cell division, become quiescent, and are "set aside" to eventually form adult NSCs (type B cells) responsible for adult neurogenesis in mouse[6–8,26]. Remaining radial glia continue to divide rapidly and contribute to embryonic neurogenesis. In the first postnatal month, rapidly dividing radial glia differentiate into neurons and glia, whereas quiescent radial glia transition into adult NSCs, ependymal cells, and parenchymal astrocytes[6–8,27]. The long radial processes retract during early postnatal development for both populations of radial glia. As radial glia transition into adult NSCs, the types of neurons they generate also change. V-SVZ radial glia produce cortical excitatory neurons and striatal neurons, whereas adult NSCs generate olfactory bulb interneurons[1,13,28]. Neurogenesis in most brain regions, except the V-SVZ-olfactory bulb and dentate gyrus, ends postnatally, as radial glia transition into adult cell types.

Similar to mouse, radial glia in humans, including outer radial glia and ventricular/truncated radial glia, are responsible for rapid embryonic neurogenesis[29,30]. Postnatally, radial glia with long radial processes disappear, and neurogenesis in most brain regions such as the cerebral cortex ends[31,32]. Thus, investigating the mechanisms that regulate postnatal changes of NSCs in mice will provide a foundation for future studies that improve our understanding of human NSC biology.

While studying the role of epigenetic regulators in astrocytes, we investigated the function of the *PR-domain containing 16* (*Prdm16*) gene. Prdm16 is an epigenetic regulator with histone methyltransferase activity[33,34]. In an experiment initially designed to study astrocyte development, we used the hGFAP-Cre transgenic mouse strain to generate *Prdm16* conditional knockout (cKO) mice. hGFAP-Cre mice express Cre recombinase in astrocytes as well as in radial glia starting from E13.5[35], precisely when quiescent radial glia are set aside to become adult B cells[6,7]. Unexpectedly, we observed the persistence of radial glia in the adult V-SVZ of *Prdm16* cKO mice and prolonged postnatal generation of neuroblasts in the cerebral cortex. To our knowledge, this is the earliest instance where radial glia are retained in adulthood in a mutant mouse. Furthermore, we found that Prdm16 deficiency led to an increase in the level of Vascular Cell Adhesion Molecule 1 (Vcam1) and that the extended presence of radial glia and neurogenesis phenotype is rescued in *Prdm16-Vcam1* double cKO mice. Our study identified Prdm16 and Vcam1 as key regulators that control the cessation of neurogenesis and the postnatal disappearance of radial glia, addressing a crucial knowledge gap in NSC biology.

## Results

### Retention of radial glial processes in adult *Prdm16* cKO mice
To investigate the potential roles of Prdm16 in the postnatal transformation of radial glia, we crossed *human-GFAP-Cre (hGFAP-Cre)* transgenic mice, which express Cre recombinase in radial glia starting

from embryonic day 13.5 (E13.5)[35] with *Prdm16*-floxed knock-in mice[36] to generate *hGFAP-Cre^{Tg/+}; Prdm16^{fl/fl}* cKO mice. We first assessed the distribution of Prdm16 protein in the postnatal brain and detected it predominantly in the V-SVZ (Supplementary Fig. 1). Prdm16 protein is reduced between E15 and P0 and remains almost undetectable afterwards in the V-SVZ of cKO mice (Supplementary Fig. 1). These *Prdm16* cKO mice are viable and fertile. As controls, we used *Prdm16^{fl/fl}* littermates without *hGFAP-Cre*.

We next assessed the presence of radial glia in postnatal *Prdm16* cKO and control mouse brains using radial glia markers GFAP and Nestin. In control mice, radial processes are abundant in neonates but mostly retract by P14. Strikingly, radial processes are still present in the V-SVZ in juvenile (one-month-old) and adult (seven-month-old) *Prdm16* cKO mice (Fig. 1a−d and Supplementary Fig. 2). We observed similar existence of radial processes in the striatum (Fig. 1a, c) and the cerebral cortex (Fig. 1b, d), where persisting radial processes are mostly present in the anterior cingulate cortex and motor cortex (Fig. 1b, d). This interesting observation suggests that radial glia may be retained in adult V-SVZ in *Prdm16* cKO mice.

### NSCs are increased and ependymal cells are decreased in *Prdm16* cKO mice
Postnatally, radial glia give rise to two major cell types, type B cells and ependymal cells, on the surface of the lateral ventricular wall in adult mice[37]. Type B cells are adult NSCs[1], whereas ependymal cells produce signals that regulate NSC proliferation and have multiple cilia that propel the circulation of cerebrospinal fluid in the ventricles[38,39]. NSCs (embryonic radial glia and adult type B cells) are GFAP+ and Vcam1+ and contain a primary cilium associated with basal bodies (labeled by γ-tubulin)[40]. Ependymal cells are Vcam1− and contain multiple γ-tubulin+ basal bodies connected to an array of motile cilia (Fig. 2b, c)[39]. If embryonic radial glia are retained in juvenile and adult *Prdm16* cKO mice, we would expect an increase in Vcam1+ cells containing a single γ-tubulin+ basal body and a corresponding decrease in Vcam1− cells containing multiple γ-tubulin+ basal bodies on the surface of the lateral ventricular wall.

To examine the cell composition of the surface of the lateral ventricular wall, we prepared whole-mount preparations using one-month-old *Prdm16* cKO and control mice to expose the surface of the lateral wall[41] and stained them with antibodies against Vcam1, γ-tubulin, and Sox2 (nuclear label of both NSCs and ependymal cells for cell counting) (Fig. 2a−c). Interestingly, *Prdm16* cKO mice showed an increase in NSCs and a decrease in ependymal cells (Fig. 2d, e). In addition, we found that *Prdm16* cKO mice often exhibit hydrocephalus, a phenotype commonly observed in mice with ependymal cell defects[42,43] (Supplementary Fig. 3). The increase in NSCs in *Prdm16* cKO mice is consistent with the hypothesis that radial glia persist postnatally in these mice.

Properties of progenitor populations in the lateral (striatum), dorsal (cortex), and medial (septum) walls of the lateral ventricle are different. Therefore, in addition to examining the cell composition in the lateral wall, we also examined NSC density in the dorsal and medial walls. We found no significant difference in the density of Pax6+ NSCs in the dorsal and medial walls between *Prdm16* cKO and control mice (Supplementary Fig. 4), in contrast to the increase in NSCs in the lateral wall in *Prdm16* cKO mice (Fig. 2d, e), demonstrating region-specific effect of Prdm16 in brain development.

### Postnatal retention of quiescent radial glia in *Prdm16* cKO mice
Having observed the presence of cells with the morphological (long radial process) and molecular (GFAP+Nestin+Vcam1+, single γ-tubulin+ basal body) features of radial glia in juvenile and adult V-SVZ in *Prdm16* cKO mice, we next utilized BrdU pulse chase to evaluate whether these cells were embryonically born radial glia persisting postnatally, or an aberrant cell type that are born postnatally.

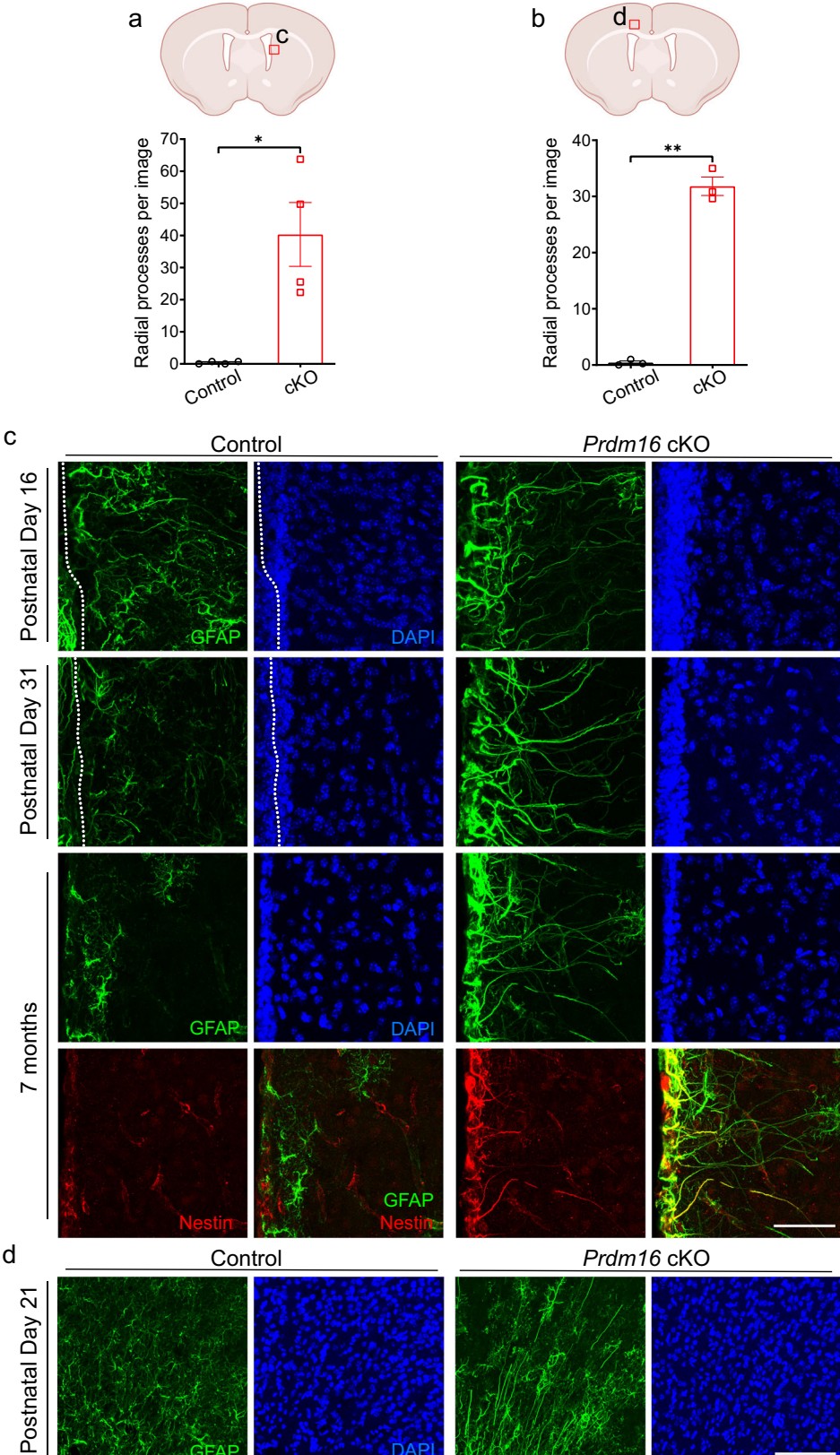

The subpopulation of radial glia set aside to become adult type B cells divide slowly whereas other embryonic radial glia divide rapidly[6,7]. Since BrdU labelling is diluted with each cell cycle in rapidly dividing radial glia and the only cells retaining high levels of BrdU several weeks after injection are slowly dividing/quiescent radial glia. We injected BrdU between E15.5 and E17.5 and

sacrificed the mice at P21 (Fig. 3a). The majority of BrdU signal colocalizes with the NSC marker GFAP (Supplementary Fig. 5). We found that the number of BrdU-labeled cells in the V-SVZ was higher in *Prdm16* cKO mice compared with control mice at P21 (Fig. 3b, c), suggesting an increase in the postnatal retention of quiescent radial glia.

**Fig. 1 | Persistence of cells with radial processes in juvenile and adult *Prdm16* cKO mice. a** A diagram shows the area of lateral V-SVZ used in the quantification and shown in (**c**). Quantification of the number of radial processes at one month of age in the striatum. Images were taken between −0.1 mm posterior and 0.74 mm anterior to Bregma. Only radial processes that extend beyond 200 µm away from the ventricular walls were counted. *n* = 4 mice per genotype. All statistical tests were performed using data from each mouse (biological replicate) as an individual observation in all figures except for scRNAseq. Two-tailed Welch's *t*-test. *p* = 0.0276. In all figures: *, *p* < 0.05. **, *p* < 0.01. ***, *p* < 0.001. N.S. not significant. Data are presented as means ± SEM as appropriate. Created in BioRender. Zhang, Y. (2025)

https://BioRender.com/y53n311. **b** A diagram shows the area of the cerebral cortex used in the quantification and shown in (**d**). Quantification of the number of radial processes at P21 in the cerebral cortex. Images were taken between 0.1 and 1.18 mm anterior to Bregma. Only radial processes that extend beyond the white matter were counted. *n* = 3 mice per genotype. Two-tailed Welch's *t*-test. *p* = 0.0021. Data are presented as means ± SEM as appropriate. Created in BioRender. Zhang, Y. (2025) https://BioRender.com/y53n311. **c** Persistence of GFAP⁺/Nestin⁺ radial processes in juvenile and adult *Prdm16* cKO mice. Scale bar: 50 µm. The dashed lines indicate ventricular walls. **d** Persistence of GFAP⁺ radial processes in P21 *Prdm16* cKO cortex. Scale bar: 100 µm.

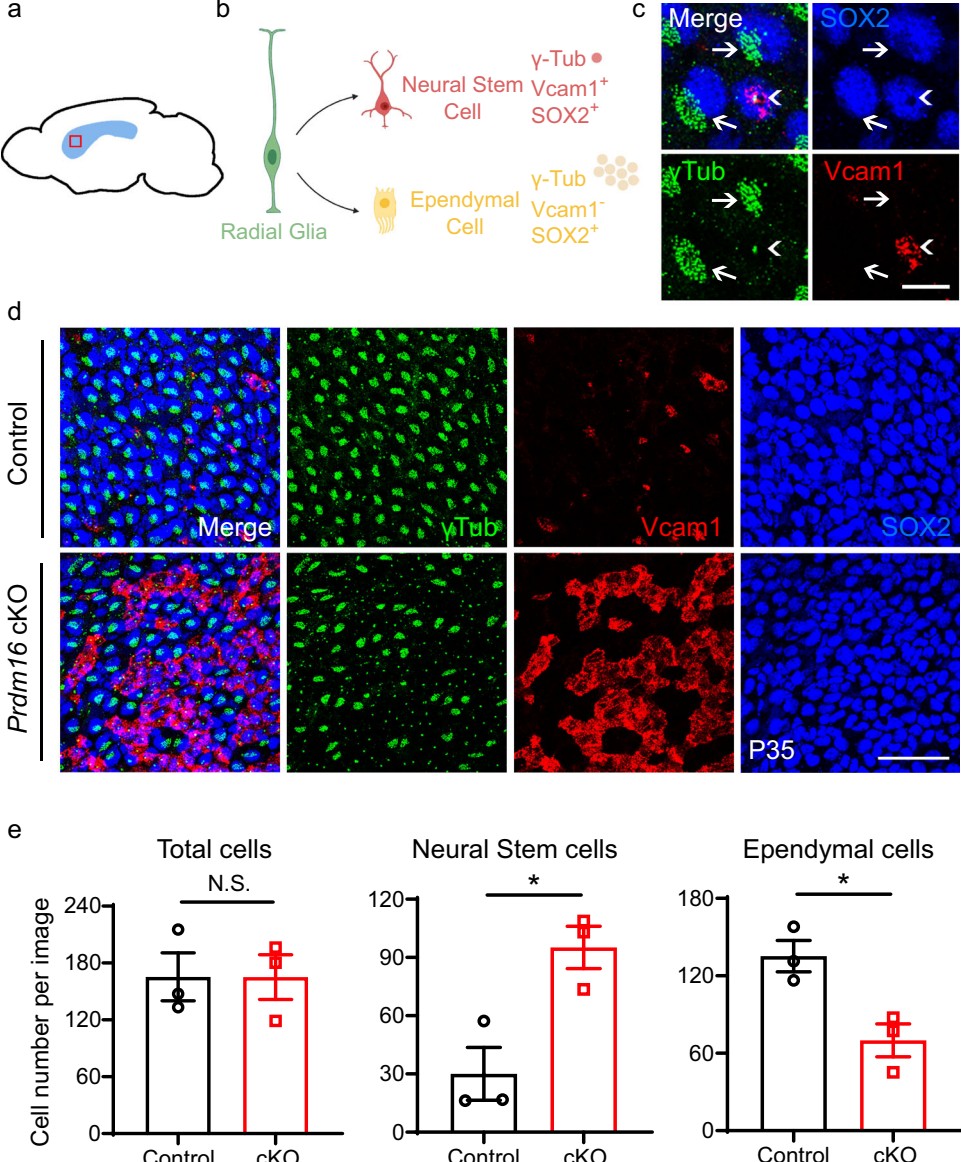

**Fig. 2 | An increase in neural stem cells and a decrease in ependymal cells in *Prdm16* cKO V-SVZ. a** Whole-mount preparation, in which the entire surface of the lateral walls of the lateral ventricles is cut out for immunohistochemistry experiments. **b** Neural stem cells and ependymal cells were identified by a combination of markers on whole-mount preparations. Neural stem cells are positive for Vcam1 and Sox2 with single dots positive for γ-tubulin immunofluorescence (a single basal body associated with a single primary cilium). Ependymal cells are negative for Vcam1 with clusters of γ-tubulin immunofluorescence (multiple basal bodies associated with multiple cilia). Created in BioRender. Zhang, Y. (2025) https://

BioRender.com/p63y337. **c** An example of a Vcam1⁺ neural stem cell with a single γ-tubulin⁺ dot (arrowhead) and examples of Vcam1⁻ ependymal cells with clusters of γ-tubulin⁺ dots (arrows). Scale bar: 10 µm. The experiment was repeated more than three times. **d** Vcam1, γ-tubulin, and Sox2 immunofluorescence on whole-mount preparations of P35 *Prdm16* cKO and control mice. Scale bar: 50 µm. **e** Quantification of neural stem and ependymal cells at P34−35. *n* = 3 mice per genotype. Two-tailed Welch's *t*-test, total cells *p* = 0.9949, neural stem cells *p* = 0.0219, ependymal cells *p* = 0.0208. Data are presented as means ± SEM as appropriate.

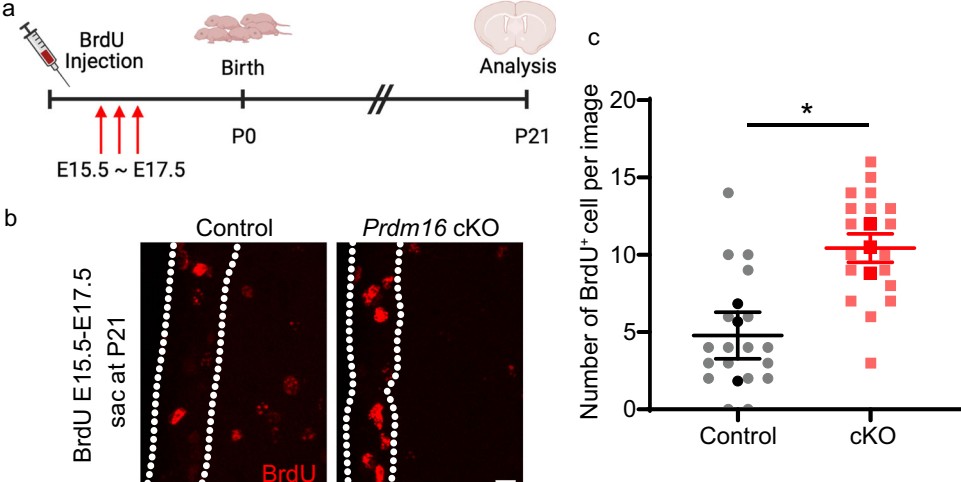

**Fig. 3 | Embryonic radial glia persist postnatally in the V-SVZs of *Prdm16* cKO mice. a** A diagram shows the design of the BrdU pulse-chase experiment. BrdU was injected 3 times daily between E15.5 and E17.5, and the mice were sacrificed and analyzed at P21. Created in BioRender. Zhang, Y. (2025) https://BioRender.com/z56b814. **b** Increased BrdU-labeled cells in the V-SVZs of *Prdm16* cKO mice compared with control at P21. The dashed lines delineate V-SVZs. Scale bar: 10 μm. **c** The quantification of BrdU-labeled cells in the V-SVZ. *n* = 3 mice per genotype. The black and red dots indicate average results from each mouse and the grey and pink dots indicate the raw counts from each image. Two-tailed Welch's *t*-test, *p* = 0.0431. Data are presented as means ± SEM as appropriate.

## Single-cell RNA sequencing (scRNA-seq) reveals molecular changes

To characterize the transcriptome profiles of postnatal NSCs in *Prdm16* cKO mice and determine whether they retain gene signature of embryonic radial glia, we performed scRNA-seq from one-month-old V-SVZ. We micro-dissected the lateral walls of the lateral ventricles using the same method described for the preparation of wholemounts[41], dissociated the tissue, and performed scRNA-seq with the 10× Genomics platform using cells from three cKO and three control mice. After filtering out low-quality cells, we obtained transcriptome data for 36,420 V-SVZ cells. We next performed unbiased cell clustering and used uniform manifold approximation and projection (UMAP) for data visualization. We annotated the identities of cell clusters based on known cell-type-specific markers (Fig. 4a, b). The sequenced V-SVZ cells included all expected cell types identified in published scRNA-seq datasets of the V-SVZ[10,44–46], such as NSCs, type C cells/transit amplifying precursors, type A cells/neuroblasts, neurons, ependymal cells, astrocytes, oligodendrocyte precursor cells (OPCs), oligodendrocytes, microglia, endothelial cells, and pericytes (Fig. 4a, b and Supplementary Fig. 6a–c). When we compared *Prdm16* cKO vs control mice, we detected differentially expressed genes in all cell types, with NSCs exhibiting the most robust transcriptome changes (Fig. 4c and Supplementary Data 1 and 2).

We next assessed whether embryonic radial glia are retained postnatally in *Prdm16* cKO mice. Because embryonic radial glia and adult NSCs share all the known NSC marker genes, the presence or absence of markers cannot distinguish radial glia from adult NSCs. However, genome-wide analyses of the *levels* of gene expression may provide insights. We therefore examined whether NSCs in juvenile *Prdm16* cKO mice retain embryonic radial glia gene signatures. For this purpose, we first mined a published embryonic and adult brain scRNA-seq dataset to identify genes differentially expressed by embryonic radial glia and adult NSCs[10]. We extracted the E14 embryonic radial glia cluster data and generated subclusters (Supplementary Figs. 7 and 8a). Based on proliferation marker genes, we identified a quiescent radial glia subcluster and an activated radial glia subcluster (Supplementary Fig. 8b–d). We then compared the gene expression of quiescent radial glia *vs.* adult type B cells and identified differentially expressed genes (RGvB-DEGs). To assess the expression of RGvB-DEGs in *Prdm16* cKO and control NSCs, we next performed gene set enrichment analysis

(GSEA). We used genes with higher expression in radial glia than adult B cells as one gene set and genes with lower expression in radial glia than adult B cells as a second gene set. Interestingly, we found that radial glia marker genes are enriched in *Prdm16* cKO NSCs compared to controls and that adult NSC-enriched genes are depleted in *Prdm16* cKO NSCs compared to controls (Fig. 4d, *p* < 0.001), demonstrating that postnatal NSCs in *Prdm16* cKO mice retain embryonic radial glia gene signatures.

Together, the three lines of evidence described above suggest that embryonic radial glia is retained postnatally in *Prdm16* cKO mice: (1) cells with the radial morphology of embryonic radial glia persist in adult V-SVZ (Fig. 1), (2) quiescent radial glia labeled with BrdU at E15.5–17.5 are increased in the V-SVZs of juvenile *Prdm16* cKO mice (Fig. 3), and (3) the transcriptome profile of juvenile *Prdm16* cKO NSCs resembles that of embryonic radial glia (Fig. 4d).

To gain insight into the function of genes differentially expressed in *Prdm16* cKO mice, we performed gene ontology (GO) and protein-protein interaction network analyses. Among the upregulated genes, 23 out of 228 are within a protein-protein interaction network involved in oxidative phosphorylation (Supplementary Fig. 9), and the GO term oxidative phosphorylation is significantly enriched among the upregulated genes (Fig. 4e and Supplementary Data 3). These genes encode mitochondrial electron transport chain components among others (e.g., *Ndufa9, Uqcr10, and Uqcrb*), suggesting that Prdm16 inhibits the expression of genes involved in oxidative phosphorylation. Our results uncover a potential role of Prdm16 in regulating energy metabolism in NSCs. GO terms enriched in genes downregulated in *Prdm16* cKO NSCs include "nervous system development" (Fig. 4e and Supplementary Data 3), consistent with a role of Prdm16 in NSC development. We also performed GSEA and obtained similar results as the GO term enrichment analysis (Supplementary Fig. 10).

To determine how Prdm16 deficiency affects cell types other than NSCs, we performed GO analysis of differentially expressed genes in each cell type (Supplementary Data 4). Similar to NSCs, oxidative phosphorylation/cell respiration genes were upregulated in transit amplifying cells, neuroblasts, astrocytes and OPC (Supplementary Fig. 11). In addition, cholesterol synthesis genes were downregulated in astrocytes (Supplementary Fig. 11c) and translation related genes were upregulated in OPCs (Supplementary Fig. 11d). These changes demonstrate that, while Prdm16 regulates oxidative phosphorylation

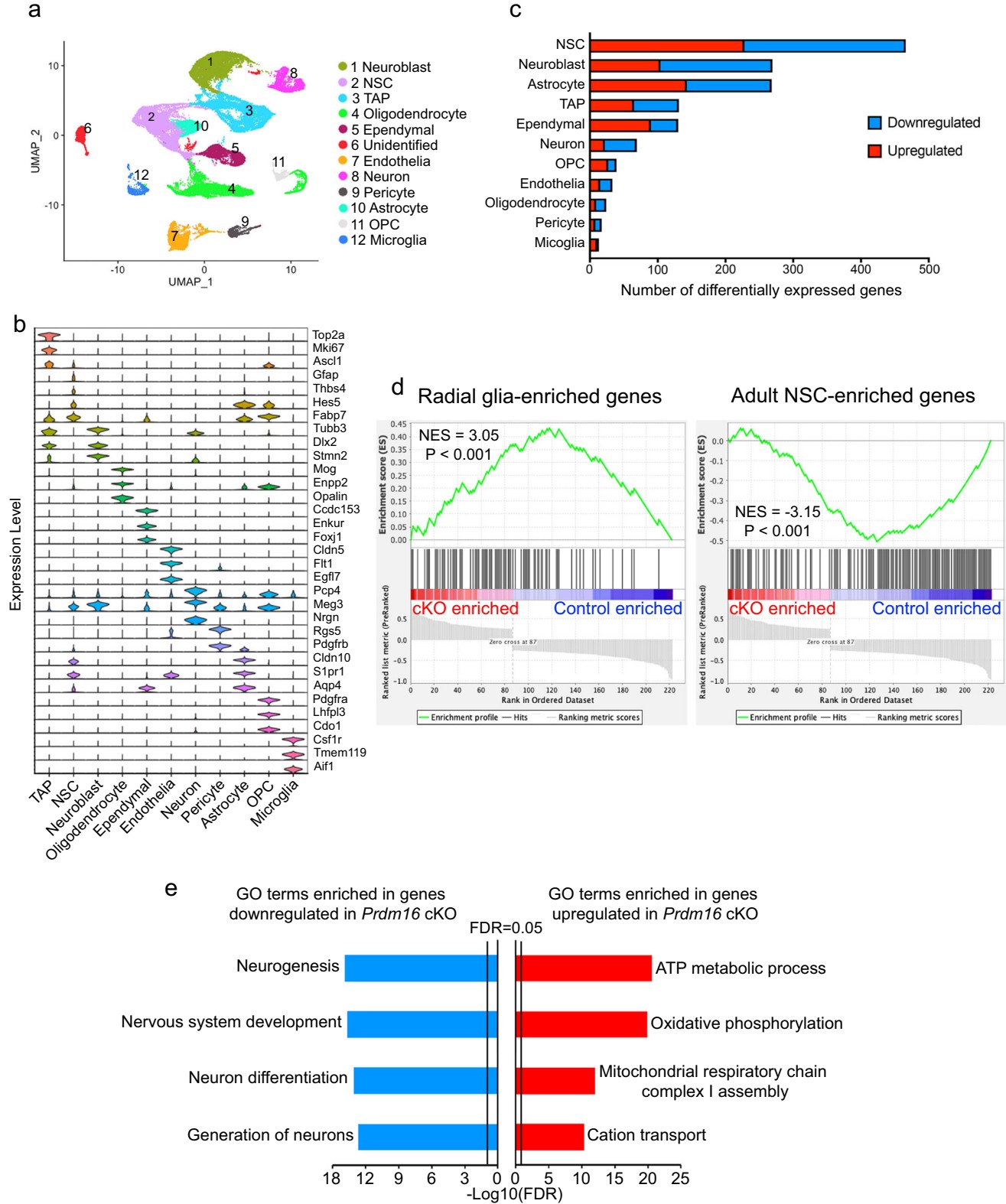

gene expression consistently across multiple cell types, it also exhibits cell type specific effects on gene expression.

## Cortical neuroblast production is prolonged postnatally

Embryonic radial glia rapidly generate millions of cortical neurons within weeks[1]. However, cortical neurogenesis ends as radial glia disappear shortly after birth[47]. This precipitous cessation of neurogenesis is accompanied by a marked reduction in the regenerative capacities of the brain. To increase the duration of neurogenesis by NSCs in stem cell-based therapies for neurological disorders, understanding the mechanisms that regulate the cessation of neurogenesis is a crucial step. However, this fundamental question remains understudied.

Having observed the postnatal persistence of radial glia in *Prdm16* cKO mice, we next examined whether these cells support the continued generation and migration of new cortical neurons by staining for doublecortin (DCX), a marker for newly generated neurons. At P21,

**Fig. 4 | ScRNAseq analysis reveals enrichment of radial glia gene signature in NSCs from juvenile *Prdm16* cKO mice. a** V-SVZs from one-month-old mice were used for scRNAseq. UMAP of scRNAseq data. Major cell clusters were identified and color-coded. **b** Expression of cell-type markers by the major cell types. **c** The numbers of differentially expressed genes in each cell type. **d** *Prdm16* cKO NSCs upregulated the expression of embryonic radial glia-enriched genes and down-regulated the expression of adult NSCs-enriched genes. NES normalized enrichment score. one-sided Kolmogorov–Smirnov (K–S)-like test. In GSEA, the enrichment score is calculated by "walking down" a ranked list of genes, adding a value to a running sum when a gene belongs to the gene set being tested, and subtracting a value when it does not; the final enrichment score is the maximum deviation from zero encountered during this process, essentially reflecting how

concentrated the gene from the gene set are at either the top or bottom of the ranked list[74,75]. In the left panel, a positive enrichment score indicates that radial glia genes are enriched in the gene set containing genes upregulated in *Prdm16* cKO mice. In the right panel, a negative enrichment score indicates that adult NSC genes are enriched in the gene set containing genes downregulated in *Prdm16* cKO mice. The black bars in the middle panel represent individual genes. In the left panel, concentration of black bars on the left indicates that radial glia genes are enriched in the gene set containing genes upregulated in *Prdm16* cKO mice. In the right panel, concentration of black bars on the right indicates that adult NSC genes are enriched in the gene set containing genes downregulated in *Prdm16* cKO mice. **e** Selected GO terms enriched in genes up and downregulated in *Prdm16* cKO NSCs compared with controls.

few DCX+ cells were detected in the cortex of control mice, consistent with prior reports[47]. However, we found that numerous DCX+ cells were present in the cerebral cortex of *Prdm16* cKO mice. These cells were concentrated in the secondary motor cortex and anterior cingulate cortex dorsal to the cingulate bundle, the area where persisting radial processes were most abundant (Fig. 5a–d). Some DCX+ cells were present along GFAP+ radial processes, possibly migrating along them (Fig. 5c). By P30, few DCX+ cells were in the cortical grey matter, but an increase in DCX+ cells in dorsal corpus callosum (the cingulate bundle area) was still observed in *Prdm16* cKO mice (Supplementary Fig. 12).

We next assess whether these cortical DCX+ cells are generated postnatally. We performed BrdU injections at P14 to label the cells generated at P14 and sacrificed the mice at P21 (Fig. 5e). By examining DCX+/BrdU+ cells in the cortex, the cells that were produced at P14 and became DCX+ newborn neurons one week later can be quantified. Interestingly, the number of DCX+/BrdU+ cells in the cortex was higher in *Prdm16* cKO mice compared with control mice at P21 (Fig. 5f, g). These results suggest that Prdm16 promotes the termination of cortical neurogenesis.

During embryonic cortical development, newly born cortical neurons mature and integrate into developing neural circuits and acquire mature neuron markers such as NeuN, Satb2, and Foxp1. Whether the cortical environment in juvenile mice can also support the maturation and integration of new neurons is unknown. To examine whether postnatally born DCX+ neuroblasts can survive and become mature neurons in the cortex, we injected 5-ethynyl-20-deoxyuridine (EdU) at P14 and followed the fate of cells born at P14 by co-labeling EdU with mature neuron markers NeuN, Satb2, and Foxp1. Very few EdU+ cells are also positive for these markers at P28 in *Prdm16* cKO or control mice (Supplementary Fig. 13), suggesting that post-natally born neurons in cKO mice do not become mature cortical neurons. They likely died after failing to integrate into neural circuits.

In the postnatal and adult brain, newly generated neurons from NSCs migrate to the olfactory bulb via the rostral migratory stream[13,48]. To examine the development of olfactory bulb inter-neurons, we performed immunostaining at P21 with antibodies against markers of two subtypes of olfactory bulb interneurons: TH+ and PV+ interneurons. Interestingly, the densities of both TH+ and PV+ interneurons were reduced in *Prdm16* cKO mice (Supplementary Fig. 14), demonstrating that the increase in cortical neurogenesis occurs potentially at the expense of olfactory bulb neurogenesis. In addition to producing neurons, radial glia also generate glial cells, including astrocytes and oligodendrocytes. We examined the densities of astrocytes, OPCs, oligodendrocytes, and microglia and did not detect significant differences between *Prdm16* cKO and control mice (Supplementary Fig. 15).

Having observed a prolonged time span of postnatal cortical neuroblast production, we next examined the proliferation of NSCs in the postnatal V-SVZ by double labeling with a proliferative cell marker Ki67 and an NSC marker Sox9. We found that the densities of Ki67+Sox9+ proliferating NSCs are reduced in *Prdm16* cKO V-SVZs compared to controls, suggesting that most NSCs in the V-SVZs of

*Prdm16* cKO mice are slowly dividing or quiescent (Supplementary Fig. 16).

## Downstream molecular mechanisms

To identify the mechanisms through which Prdm16 regulates the postnatal disappearance of embryonic radial glia, we took a candidate-based approach. We initially used Vcam1 as a NSC marker and observed an increase in Vcam1 proteins in the V-SVZ of *Prdm16* cKO mice (Fig. 2d) using whole-mount en-face preparations and further confirmed this finding using coronal brain sections (Fig. 6a, d). Interestingly, we noticed that the increase in the level of Vcam1 protein immunofluorescence (Figs. 6a, d and 2d) is far greater than the increase in the number of NSCs (Fig. 2e), suggesting that the levels of Vcam1 protein in each NSC is elevated in *Prdm16* cKO mice. Therefore, Prdm16 normally acts to repress Vcam1 and its absence thus leads to a de-repression and an increase in Vcam1. Vcam1 levels are high in embryonic radial glia and decrease postnatally (Supplementary Fig. 2 and previous report[49]). Importantly, Vcam1 deficiency results in a premature depletion of NSCs[49,50], which contrasts with the postnatal radial glia retention phenotype observed in *Prdm16* cKO mice. These results led us to hypothesize that high Vcam1 levels during the embryonic stage maintains radial glia and that Prdm16 induces a postnatal reduction in Vcam1, leading to postnatal disappearance of radial glia and cessation of cortical neurogenesis. If this hypothesis is true, the developmental timer operates as an hourglass, where the gradual loss of Vcam1 in postnatal NSCs (Supplementary Fig. 2a) is similar to the loss of sand from the top of the hourglass and serves as a measure of time.

To test this hypothesis, we generated *Prdm16-Vcam1* double cKO (hGFAP-Cre[Tg/+]; *Prdm16*[fl/fl]; *Vcam1*[fl/fl], DKO) mice. Interestingly, the radial glia retention phenotype is rescued in DKO mice (Fig. 6b, e), suggesting that Prdm16 promotes the retraction of radial glia basal processes by inhibiting Vcam1. We next examined wholemount en-face preparations of the V-SVZ of one-month-old DKO mice to assess the abundance of NSCs. We found that the densities of NSCs and epen-dymal cells were restored to control levels in *Prdm16-Vcam1* DKO mice (Fig. 6c, f), again supporting the hypothesis that Prdm16 promotes the postnatal transformation of radial glia by repressing Vcam1.

We further examined whether Prdm16's repression of Vcam1 is involved in the cessation of cortical neurogenesis. We found that the prolonged postnatal neurogenesis phenotype observed in *Prdm16* cKO mice was completely rescued in *Prdm16-Vcam1* DKO mice (Fig. 6g, h), suggesting that the elevation of Vcam1 levels in *Prdm16* cKO mice is essential for the postnatal continuation of cortical neurogenesis.

These results support an hourglass model of the developmental timer: the gradual loss of Vcam1 in postnatal NSCs (Supplementary Fig. 2) resembles sand flowing from the top of the hourglass, serving as a measure of time. When the top of the hourglass runs out of sand (a low Vcam1 level is reached), cortical neurogenesis terminates. Prdm16 starts the running of sand in the hourglass by repressing Vcam1. In *Prdm16* cKO mice, Vcam1 stays high (Fig. 6a, d and Supplementary Fig. 2), the sand does not run down from the top of the

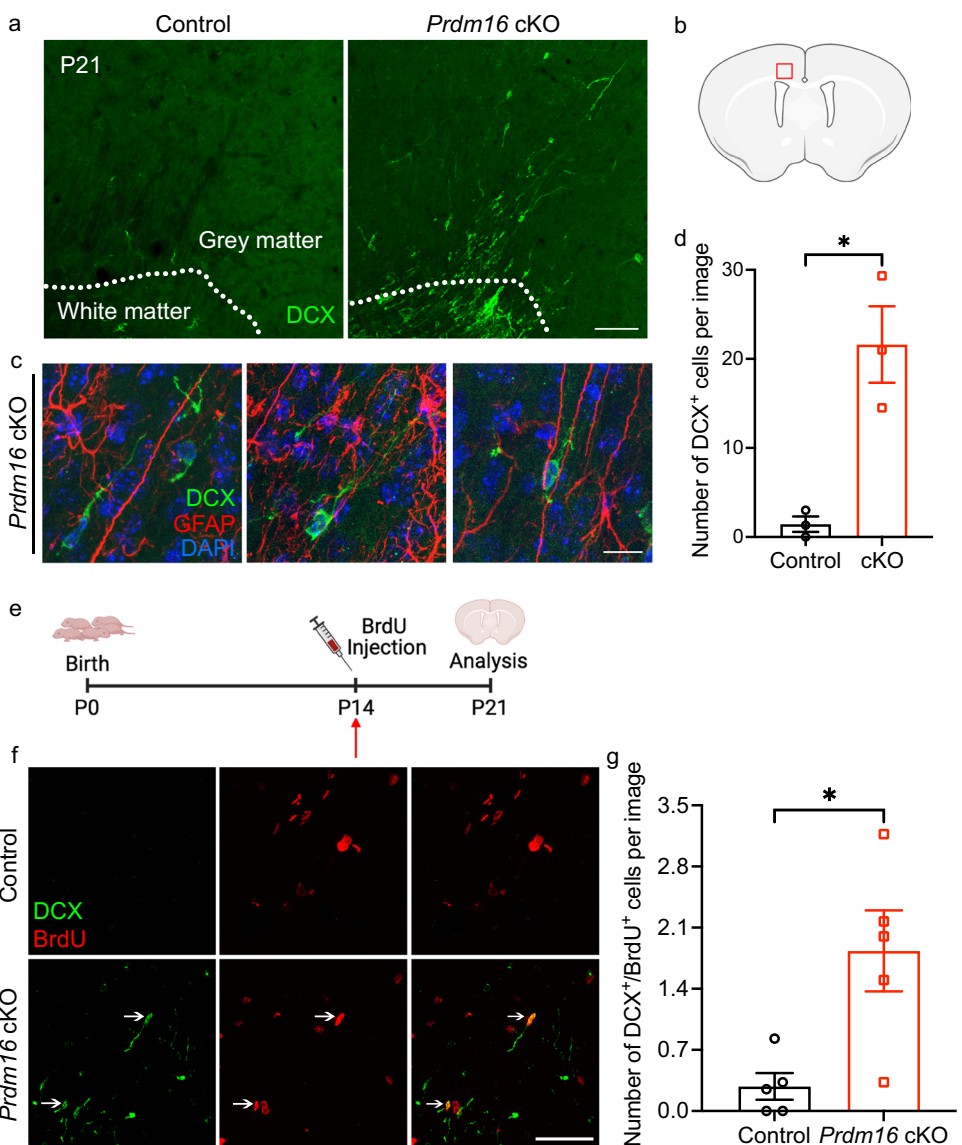

**Fig. 5 | Postnatal persistence of cortical neurogenesis in *Prdm16* cKO mice.**
**a** Numerous DCX⁺ newly born neurons in *Prdm16* cKO cerebral cortices at P21. Few new neurons are detected in control cerebral cortex at this age. The dashed lines delineate the boundaries between the white matter and the grey matter. Scale bar: 100 μm. The experiment was repeated more than three times. **b** A diagram shows the area shown in (**a**) and (**c**) and quantified in (**d**). Created in BioRender. Zhang, Y. (2025) https://BioRender.com/y53n31l. **c** DCX⁺ newborn neurons alongside GFAP⁺ radial processes in the cerebral cortex of *Prdm16* cKO mice at P21. Scale bar: 20 μm. **d** Quantification of the number of DCX⁺ cells at P21 in the cortex. Only DCX⁺ cells in the grey matter were counted. $n = 3$ mice per genotype. Two-tailed Welch's *t*-test.

$p = 0.0379$. Data are presented as means ± SEM as appropriate. **e** A diagram shows the design of the postnatal BrdU pulse-chase experiment. BrdU was injected at P14, and the mice were sacrificed and analyzed at P21. Created in BioRender. Zhang, Y. (2025) https://BioRender.com/r15n645. **f** Increased DCX⁺/BrdU⁺ cells in the *Prdm16* cKO cortex compared with control at P21. The arrows delineate DCX⁺/BrdU⁺ cells in the *Prdm16* cKO cortex. Scale bar: 50 μm. **g** Quantification of DCX⁺/BrdU⁺ cells in the cortex. Only DCX⁺/BrdU⁺ cells in the grey matter were counted. $n = 5$ mice per genotype. Two-tailed Welch's *t*-test, $p = 0.0256$. Data are presented as means ± SEM as appropriate.

hourglass, thus radial glia and cortical neurogenesis persists postnatally. In *Prdm16-Vcam1* DKO mice, the top of the hourglass is empty, Vcam1 level is low, and neurogenesis terminates postnatally.

## Cellular and molecular pathways regulated by Prdm16 and Vcam1

Given the Vcam1 rescue results (Fig. 6b, c, e–h), and the potential role of Prdm16 as an epigenetic regulator that regulates gene expression, we examined whether Prdm16 regulates Vcam1 transcription by scRNA-seq. *Vcam1* mRNA levels, however, do not differ between *Prdm16* cKO and control mice (Fig. 6k). We further utilized an orthogonal approach, RNAscope, to assess the levels of *Vcam1* mRNA and again did not detect significant differences between *Prdm16* cKO and

control mice (Fig. 6i, j). The elevation in Vcam1 protein levels without accompanying changes in mRNA levels in *Prdm16* cKO mice suggests that Prdm16 regulates Vcam1 at the post-transcriptional level, either through regulating protein synthesis or degradation. These results reveal a mechanism of Vcam1 regulation in NSCs.

To identify the cellular and molecular pathways directly regulated by Prdm16, we analyzed our scRNA-seq dataset alongside a previously published chromatin immunoprecipitation sequencing (ChIP-seq) dataset of *Prdm16* KO mice[51]. We identified direct targets of Prdm16 as genes that have Prdm16 binding peaks in the ChIP-seq dataset and are differentially expressed between *Prdm16* cKO and control NSCs in our scRNA-seq dataset. We then performed gene ontology analysis on these direct target genes and found that the top enriched term among

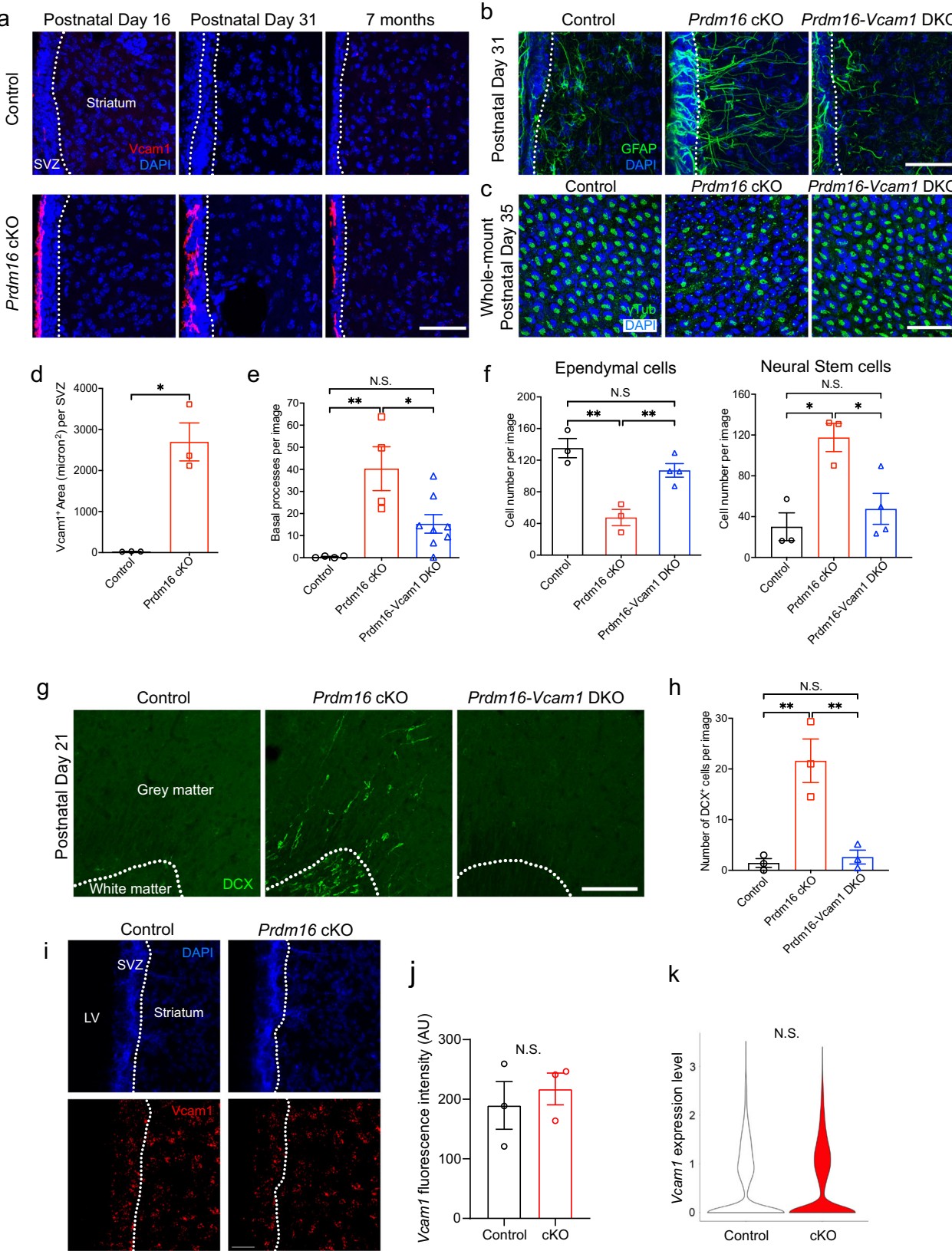

genes upregulated in *Prdm16* cKO mice is "cell junction", while the top enriched term among genes downregulated is "differentiation" (Supplementary Fig. 17). This result suggests that Prdm16 regulates NSC behavior by directly enhancing the expression of genes involved in differentiation while repressing the expression of genes encoding cell junction proteins (Supplementary Fig. 18). Notably, no Prdm16 binding

peak was found in the regulatory sequence of Vcam1 in the ChIP-seq dataset[51], confirming our hypothesis that Prdm16 does not directly regulate Vcam1 gene expression.

Our results show that Vcam1 levels are elevated in *Prdm16* cKO mice (Fig. 6a, d and Supplementary Fig. 2) and that the NSC phenotypes in *Prdm16* cKO mice are rescued in *Prdm16-Vcam1* DKO mice

**Fig. 6 | Rescue of the postnatal persistence of radial glia and cortical neurogenesis phenotypes in *Prdm16-Vcam1* DKO mice. a** Increased Vcam1 immunofluorescence in the V-SVZ of juvenile and adult *Prdm16* cKO mice. The dashed lines delineate the boundaries between the V-SVZ and the striatum and the boundary between the V-SVZ and the ventricle. Scale bar: 50 μm. **b** The persistence of GFAP⁺ radial processes is partially rescued in *Prdm16-Vcam1* DKO mice. Scale bar: 50 μm. **c** Whole-mount en-face preparations showing neural stem cells with single γ-tubulin+ dots and ependymal cells with clusters of γ-tubulin+ dots. Scale bar: 50 μm. The cell composition defect in *Prdm16* cKO mice is rescued in *Prdm16-Vcam1* DKO. **d** Quantification of Vcam1 immunofluorescence positive area in V-SVZ at P31. *n* = 3 mice per genotype. Two-tailed Welch's *t*-test, *p* = 0.0287. Data are presented as means ± SEM as appropriate. **e** Quantification of the number of radial processes at one month of age. Control, *n* = 4; cKO, *n* = 4; DCK, *n* = 8 mice. One-way ANOVA with Tukey's multiple comparisons test, control vs. *Prdm16* cKO, *p* = 0.002, *Prdm16* cKO vs. *Prdm16-Vcam1* DKO, *p* = 0.019. Data are presented as means ± SEM as appropriate. **f** Quantification of neural stem and ependymal cells at P34–35. Control, *n* = 3; cKO, *n* = 3; DCK, *n* = 4 mice. One-way ANOVA with Tukey's multiple

comparisons test. Ependymal cells, control vs. *Prdm16* cKO, *p* = 0.0017, *Prdm16* cKO vs. *Prdm16-Vcam1* DKO, *p* = 0.0097. Neural stem cells, control vs. *Prdm16* cKO, *p* = 0.0124. *Prdm16* cKO vs. *Prdm16-Vcam1* DKO, *p* = 0.0262. Data are presented as means ± SEM as appropriate. **g** The postnatal persistence of newly born DCX⁺ neurons in the cerebral cortex seen in *Prdm16* cKO mice is rescued in *Prdm16-Vcam1* DKO mice. The dashed lines delineate the boundaries between the white matter and the grey matter. Scale bar: 100 μm. **h** Quantification of DCX⁺ cells at P21 in the cerebral cortex. Only DCX⁺ cells in the grey matter were counted. *n* = 3 mice per genotype. One-way ANOVA with Tukey's multiple comparisons test. Control vs. *Prdm16* cKO, *p* = 0.0041. *Prdm16* cKO vs. *Prdm16-Vcam1* DKO, *p* = 0.0055. Data are presented as means ± SEM as appropriate. **i** *Vcam1* mRNA detected by RNAscope at P30. LV lateral ventricle. Scale bar: 50 μm. **j** Quantification of *Vcam1* mRNA levels detected by RNAscope in V-SVZ at P30. *n* = 3 mice per genotype. Two-tailed Welch's *t*-test. *p* = 0.6004. Data are presented as means ± SEM as appropriate. **k** *Vcam1* mRNA expression level in the NSCs of P34–35 *Prdm16* cKO and control mice as determined by scRNA-seq. Wilcoxon test with Bonferroni correction, N.S. not significant.

(Fig. 6a–h), suggesting that Vcam1 acts downstream of Prdm16. To understand how Vcam1 affects the expression of other genes downstream of Prdm16, we compared our scRNA-seq data from *Prdm16* cKO, *Prdm16-Vcam1* DKO, and control mice to identify Vcam1-dependent and Vcam1-independent genes. For this analysis, we focused on genes indirectly regulated by Prdm16 (those without Prdm16 binding peaks in the ChIP-seq). We identified Vcam1-dependent genes as those whose differential expression in *Prdm16* cKO versus control mice was rescued in *Prdm16-Vcam1* DKO mice (i.e., not significantly different between DKO and control). Gene ontology analysis revealed "mRNA processing" and "trans-synaptic signaling" among the top terms enriched in Vcam1-dependent genes (Supplementary Fig. 19). Additionally, we analyzed Vcam1-independent genes (those differentially expressed between *Prdm16* cKO and control mice and also differentially expressed between *Prdm16-Vcam1* DKO and control mice in the same direction) and found "ATP synthesis" as the top enriched term among these genes (Supplementary Fig. 20).

In summary, the gene regulatory network analysis indicates that Prdm16 directly enhances the expression of differentiation genes and represses genes encoding cell junction proteins. Beyond regulating these direct target genes, Prdm16 indirectly upregulates growth factor response genes in a Vcam1-dependent manner and downregulates genes involved in ATP synthesis in a Vcam1-independent manner. These analyses defined the cellular and molecular pathways regulated by Prdm16 and Vcam1 in NSCs and reveal numerous candidate genes for future studies, expanding our understanding of the termination of neurogenesis.

Based on the results described above, we propose the following working model (Fig. 7a, b): Vcam1 protein levels are high in embryonic NSCs/radial glia. Prdm16 indirectly induces a reduction in Vcam1 level postnatally, triggering the retraction of radial processes, the transformation of embryonic radial glia into postnatal cell types such as adult NSCs and ependymal cells, and the cessation of cortical neurogenesis. In *Prdm16* cKO mice, elevated Vcam1 levels drive the postnatal retention of embryonic quiescent radial glia, which slowly divide and continue to produce cortical neurons for a longer period postnatally. The postnatal retention of embryonic radial glia impairs the production of ependymal cells and olfactory bulb interneurons. In *Prdm16-Vcam1* DKO mice, Vcam1 is absent, and the transition can occur. Vcam1 levels, post-transcriptionally regulated by Prdm16, thus determine whether radial glia stay in the embryonic radial glia stage or progress to the adult NSC stage and whether cortical neurogenesis continues or ends. These results fill a crucial knowledge gap in NSC biology and may inform future studies directed toward improving NSC-based therapies by extending the time span of neurogenesis.

## Discussion

In this study, we identified roles for Prdm16 and Vcam1 in promoting the postnatal transformation of radial glia and the termination of cortical neurogenesis. These results may offer possible explanations for the differences in neurogenesis among vertebrate species. For example, fish display more prominent and widespread adult neurogenesis than mammals[52], and the changes in evolution that led to this difference have not been identified. Intriguingly, *Prdm16*, while expressed in mice after birth, is not expressed post egg hatching in zebrafish[53]. The postnatal expression of Prdm16 in mice contributes to the disappearance of radial glia and the cessation of cortical neurogenesis. On the contrary, zebrafish, which lack post-hatch Prdm16 expression, retain their radial glia and continue to generate new neurons during their adulthood[54]. These observations are consistent with an evolutionarily conserved role of Prdm16 in the postnatal disappearance of radial glia. It is likely that the postnatal expression of Prdm16 first occurred in a species that appeared later than fish and earlier than mammals, leading to the retraction of the radial processes of embryonic NSC/radial glia, precluding postnatal cortical neurogenesis and the radial migration of new cortical neurons. Thus, adult neurogenesis is more limited in mammals than in fish.

Virtually nothing is known about the mechanisms regulating the postnatal disappearance of radial glia and the termination of neurogenesis. To our knowledge, *Prdm16* cKO mice represent a mutant mouse model in which V-SVZ radial glia persist into adulthood. Building on this finding, our study uncovered Prdm16 as a window to understand the molecular logic underlying the termination of neurogenesis, with implications extending beyond a better understanding of Prdm16's specific function. Previous studies have investigated Prdm16's role in NSCs[51,55–59], particularly in upper-layer neuron development, but none have examined its specific role in the postnatal transitions of NSCs. One study reported a decrease in neurosphere-forming neural stem/progenitor cells in *Nestin-Cre; Prdm16^{fl/fl}* cKO mice[56]. In contrast, we found that quiescent NSCs are increased, and neurogenesis is extended in *hGFAP-Cre; Prdm16^{fl/fl}* cKO mice (Figs. 2 and 5). The difference in these observations could be attributed to the stage-specific activation of Nestin-Cre and hGFAP-Cre at E10.5 and E13.5, respectively[35,60], or distinct effects of Prdm16 on quiescent vs. activated NSCs. Given that Prdm16 is a histone methyltransferase, it could regulate different target genes in a developmental stage and cell type-specific manner. Future studies using CreER strains to examine the temporal requirement of Prdm16 function could shed more light on the intricacies of Prdm16's function.

Our BrdU/EdU pulse-chase experiments demonstrate that while Dcx+ neuroblasts are continuously produced in the postnatal cortex, these newly formed cells fail to become mature neurons in *Prdm16* cKO mice. Previous studies have shown that successful synapse formation

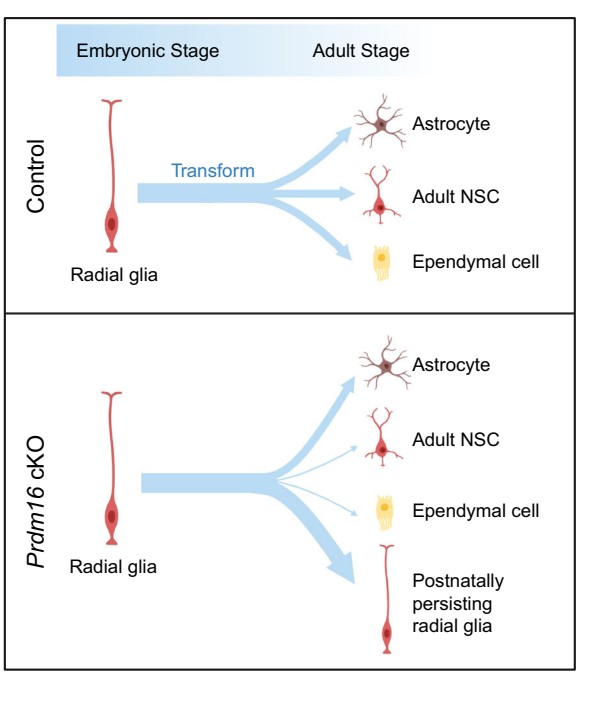

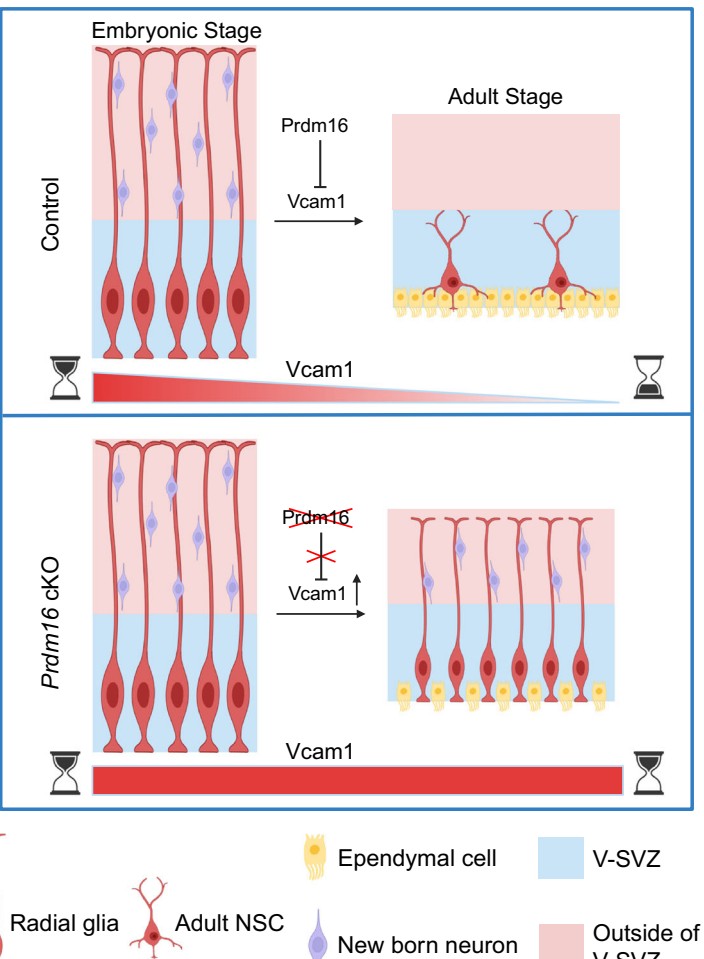

**Fig. 7 | A working model of the roles of Prdm16 and Vcam1 cessation of neurogenesis and postnatal disappearance of embryonic radial glia.**
**a** Differentiation of radial glial cells in control and *Prdm16* cKO mice. **b** A diagram shows embryonic slowly dividing radial glia become the adult type B cells postnatally. Vcam1 is highly expressed in the embryonic V-SVZ. In control mice, Prdm16 induces a reduction in Vcam1 levels postnatally, triggering the transition from embryonic radial glia to adult NSCs. In *Prdm16* cKO mice, the expression of Vcam1 stays high postnatally, leading to the persistence of radial glia and the continuation of cortical neurogenesis. Created in BioRender. Zhang, Y. (2025) https://BioRender.com/r64e200.

and circuit integration are essential for the survival of newborn neurons[61]. However, postnatal cortical circuits are largely established and offer a less supportive environment for new synapse formation compared to the embryonic period. Consequently, it is likely that these postnatally generated neurons do not survive due to their inability to integrate into the existing circuitry.

Our study reveals that increased Vcam1 levels in *Prdm16* cKO mice play a crucial role in preserving quiescent embryonic NSCs into adulthood (Fig. 6). In the aging brain, quiescent NSCs gradually deplete, resulting in decreased neurogenesis and diminished regenerative potential. By manipulating Vcam1 levels, we could potentially improve the retention of quiescent NSCs and enhance regenerative capacities in conditions of aging and neurological disorders.

While Prdm16 is essential for the postnatal disappearance of radial glia and the termination of cortical neurogenesis, our analysis of scRNA-seq datasets[10] shows no significant differences in its mRNA levels between the embryonic and postnatal stages. Hence, the exact reason why Prdm16 instigates the postnatal transitions specifically at the early postnatal stage, rather than the embryonic stage, remains elusive. The identification of other factors that interact with the Prdm16-Vcam1 pathway in the postnatal transition process in future investigations will

provide clarity on this issue. As previously noted, Prdm16 appears to impact the levels of the Vcam1 protein rather than directly regulating its transcript levels. Based on scRNAseq and ChIP-seq analyses, the regulatory region of the epidermal growth factor receptor (EGFR) gene exhibits a Prdm16 binding peak, and its expression is upregulated in NSCs from *Prdm16* cKO mice. Moreover, activation of EGFR signaling has been shown to elevate Vcam1 levels in cancer cells[62]. These findings suggest that Prdm16 may indirectly regulate Vcam1 levels through the direct transcriptional regulation of EGFR. Future research aimed at deciphering how Prdm16 indirectly influences Vcam1 protein levels will enhance our understanding of the postnatal transition process. Vcam1 is a cell adhesion molecule that can bind Integrin α4β1. Whether the adhesion function of Vcam1 is required and what downstream molecules are involved in Vcam1 function for maintaining radial glia are aspects yet to be explored in future studies.

Our scRNA-seq analyses reveal that genes associated with oxidative phosphorylation are upregulated in Prdm16-deficient NSCs, transit amplifying cells, neuroblasts, astrocytes, and OPCs, indicating a role for Prdm16 in the regulation of metabolic pathways. Interestingly, a study in acute myeloid leukemia cells demonstrated that Prdm16 overexpression activates oxidative phosphorylation[63]. These findings

suggest that Prdm16 may regulate energy metabolism in a tissue-specific manner, potentially by modulating the expression of distinct target genes across different cellular contexts.

The downregulation of cholesterol synthesis genes in astrocytes from Prdm16-deficient mice highlights a potentially significant role for Prdm16 in regulating lipid metabolism within the CNS. Cholesterol is an essential component of cellular membranes and a critical substrate for myelination and synapse formation in the brain. Astrocytes are a primary source of cholesterol in the CNS[64,65]. The disruption of cholesterol biosynthesis pathways in astrocytes could have profound implications for neuronal function and CNS homeostasis, as neurons rely on astrocyte-derived cholesterol for optimal synaptic activity and plasticity[66]. Moreover, impaired cholesterol metabolism in the CNS has been associated with neurodegenerative diseases, such as Alzheimer's disease, underscoring the importance of astrocyte-specific cholesterol production in maintaining brain health[67].

In summary, we identified Prdm16 and Vcam1 as molecules that promote the postnatal disappearance of radial glia and the cessation of cortical neurogenesis. Our study sheds new light on a poorly understood stage of NSC development and may contribute to the improvement of NSC-based therapies for neurological disorders.

## Methods

All animal experimental procedures were approved by the Chancellor's Animal Research Committee at the University of California, Los Angeles, and conducted in accordance with applicable national and state laws and policies.

### Experimental animals

All experimental animal procedures (protocols: #B-16–079 and #R-16–080) were approved by the Chancellor's Animal Research Committee at the University of California, Los Angeles and conducted in compliance with national and state laws and policies. We used mice group-housed in standard cages (2–3 adult mice, or 1–2 adults with a litter of pups per cage). Rooms were maintained on a 12-h light/dark cycle. Euthanasia was performed using gradual filling with carbon dioxide at a 60% fill rate during the light cycle. We used the following mouse strains; *hGFAP-Cre* (JAX004600), *Prdm16*-floxed (Spiegelman lab), and *Vcam1*-floxed (JAX007665). We crossed the strains to obtain control (*Prdm16*$^{fl/fl}$), *Prdm16* single knockout (*hGFAP-Cre*$^{+/-}$; *Prdm16*$^{fl/fl}$), and *Prdm16-Vcam1* double knockout mice (*hGFAP-Cre*$^{+/-}$; *Prdm16*$^{fl/fl}$; *Vcam1*$^{fl/fl}$). We also examined *Prdm16* heterozygous mice (*hGFAP-Cre*$^{+/-}$; *Prdm16*$^{fl/+}$) and did not observe any defect. We used both male and female mice in approximately equal proportions in the experiments and recorded data from male and female mice separately. Approximately equal numbers of males and females were used. We did not observe any sex-dependent effect in any assays and therefore combined data from both sexes.

### Immunohistochemistry on coronal brain sections

Mice were anesthetized with isoflurane and transcardially perfused with saline followed by 4% PFA. Brains were removed and further fixed in 4% PFA at 4 °C overnight. The brains were washed with PBS and cryoprotected in 30% sucrose at 4 °C for two days before being immersed in OCT (Fisher, cat#23-730-571) and stored at −80 °C. Brains were sectioned on a cryostat (Leica) and 30 μm floating coronal sections were blocked and permeabilized in 10% donkey serum with 0.2% Triton X-100 in PBS and then stained with primary antibodies against PRDM16 (Patrick Seale lab, dilution 1:100), PRDM16 (R&D system, AF6295-SP, 1:200), GFAP (Biolegend, 829401, 1:1500 and Dako, Z0334, 1:1500), Nestin (Aveslabs, NES-0020, 1:500), Vcam1 (BD Biosciences, 550547,1:100), DCX (Abcam, ab18723, 1:500), BrdU (Abcam, ab6326, 1:500), Parvalbumin (Novus Biologicals, NB120-11427SS, 1:500), TH (Sigma, MAB318, 1:400), Calretinin (Sigma, ZMS1073-25UL, 1:400), NeuN (AbCam, ab177487, 1:1000), Satb2 (AbCam, ab34735, 1:1000), and Foxp1

(Bennett Novitch, MI 585, 1:16000) at 4 °C overnight. Sections were washed three times with PBS and incubated for 2 h at room temperature with secondary antibodies followed by three additional PBS washes. Secondary antibodies: Donkey anti-rabbit 594 (Life Technologies, A21207, 1:1000), Donkey anti-guinea pig 594 (Jackson ImmunoResearch, 706-585-148, 1:1000), Donkey anti-rabbit 647 (Life Technologies, A31573, 1:1000). The sections were then mounted on Superfrost Plus micro slides (Fisher, cat#12-550-15) and covered with mounting medium (Fisher, cat#H1400NB) and glass coverslips. The coronal brain sections were imaged with a Zeiss widefield fluorescence microscope or a Zeiss LSM800 confocal microscope with 20, 40, and 63× lenses.

### Immunohistochemistry on the SVZ whole-mount preparations

We prepared the V-SVZ whole-mount preparations and performed immunostaining according to published protocols[41]. Briefly, we euthanized the mice and dissected out the brains. We then carefully expose the lateral walls of the lateral ventricles and cut away other brain structures. We then fixed the whole-mounts, ventricle side up, with 4% PFA with 0.1% Triton-X100 overnight at 4 °C. We then washed the whole-mounts 3 times with PBS with 0.1% Triton-X100, blocked in 10% normal donkey serum in PBS with 2% Triton-X100 at room temperature for 1 h, and stained with primary antibodies against Vcam1 (BD Biosciences, 550547, 1:100), γ-Tubulin (Sigma, T5192-25UL, 1:1000), and SOX2 (R&D Systems, AF2018, 1:1000) for 48 h at 4 °C. We then quickly rinsed the whole-mounts with PBS three times, stained them with fluorescent secondary antibodies for 48 h at 4 °C, and quickly rinsed them with PBS three times. After staining, we sub-dissected the whole-mounts to preserve only the lateral wall of the lateral ventricle as a sliver of tissue 200–300 μm thick and mounted the tissue in FluorSave™ Reagent mounting medium on glass slides with glass coverslips. We imaged the whole-mounts with a Zeiss LSM800 confocal microscope with a 40× lens.

### BrdU pulse chase

We intraperitoneally injected 50 mg/kg BrdU into pregnant female mice daily for three consecutive days at E15.5–17.5. The progeny was sacrificed by transcardial perfusion at P21. For postnatal BrdU pulse chase, we intraperitoneally injected 50 mg/kg BrdU into P14 mice, and sacrificed mice and analyzed brains at P21. The brains were processed for immunohistochemistry as described above with the following modifications: the brain sections were first treated with 2 M HCl for 30 min at 37 °C followed by treatment with 0.1 M boric acid for 10 min before adding the primary antibodies.

### Sample and library preparation for scRNA-seq

The V-SVZ of the lateral walls of the lateral ventricles from one-month-old (P31–33) mice were dissected according to published protocols[41]. The V-SVZs from 2 to 3 mice, including both males and females were combined as a biological replicate. Single-cell suspensions were prepared as described in ref. [68]. Briefly, a 2 mg/ml papain solution was pre-activated at 37 °C for 20 min. The V-SVZ samples were then digested in the papain solution at 30 °C for 30 min with a shaking speed of 60 rpm in a shaker. The tissue sample is then mechanically dissociated by trituration with a Pasteur pipette 10–15 times per round for three rounds, washed, and centrifuged with an OptiPrep density gradient to remove myelin and cellular debris. After centrifugation, the dense white layer and all layers above were carefully aspirated. The lower layers were pelleted at $200 \times g$ for 3 min, resuspended in 1 ml PBS with 0.04% bovine serum albumin (0.04% BSA-PBS), filtered through 40 μm strainers, and resuspended in 0.04% BSA-PBS at 700–1200 cells/μl for single-cell capture. Cell viability was assessed by Trypan Blue staining and a final cell count was performed before single cell capture. The single-cell suspensions with > 60% viable cells were used for single-cell capture. The 10× Genomics Chromium Next GEM Single Cell 3′ Reagent Kits v3.1 were used for GEM generation, barcoding, and library preparation according

to the manufacturer's instructions. Reagents were prepared to aim to capture 7000–10,000 cells per sample. The TapeStation was used for cDNA and library quality control. The libraries were sequenced with an Illumina NovaSeq sequencer with S2 ×100 cycles at the UCLA Technology Center for Genomics and Bioinformatics to obtain 90,871 reads per cell on average.

### scRNA-seq data analysis

Reads were mapped to the mouse genome (GRCm38) with Cellranger Count (version 6.1.2). Feature-count matrices generated by Cell Ranger were next processed with Seurat version 4.1.0[69] for filtering, clustering, and differential gene expression analysis. Low-quality cells with fewer than 500 features or higher than 10% mitochondrial genes were filtered out. Potential doublets with more than 7000 features were also removed. In total, we obtained transcriptome data for 36,420 V-SVZ cells that passed quality control, including 21,939, 14,481, and 6886 cells from control, *Prdm16* cKO, and *Prdm16-Vcam1* DKO mice, respectively. The data were then normalized and scaled using the "LogNormlize" method in the Seurat package. We then found the top 2000 highly variable features and used the top 30 principal components for dimensional reduction. We used the FindIntegrationAnchors function in Seurat to anchor and integrate individual samples together and removed batch effects. We next clustered the cells using 0.15 resolution and used UMAP for dimension reduction. We assigned cell type identity to clusters based on known cell-type specific marker genes[10,45,70–72] and identified differentially expressed genes using the FindMarkers function in Seurat. To calculate pseudobulk gene expression for each cell type, we used the AverageExpression function in Seurat. For the analysis of the E14.5 dataset[10], we first integrated data from all E14.5 samples using the IntegrateData function in Seurat. We then removed low-quality cells with fewer than 500 features or higher than 6% mitochondrial genes, and potential doublets with more than 6000 features. Next, we identified the NSC cluster based on the expression of *Nestin, Fabp7*, and *Vcam1*. For GO and protein-protein interaction network analysis, we used string-db.org with default settings[73].

### Gene set enrichment analysis (GSEA)

To confirm the GO analysis results, we also performed Gene Set Enrichment Analysis. We downloaded GSEA (version 4.2.3) software from www.gsea-msigdb.org. We used the default settings with the following exceptions: we first ranked the *Prdm16* cKO NSC *vs.* control NSC differentially expressed genes by fold change, then ran GSEA-Preranked with this ranked gene list. We used c5.go.bp.v7.5.1.symbols.gmt as "Gene sets database". We obtained similar results by GSEA as the GO term enrichment analysis.

To compare *Prdm16* cKO NSC *vs.* control NSC differentially expressed genes and slowly dividing radial glia *vs.* adult NSC differentially expressed genes by GSEA, we built our own gene sets of slowly dividing radial glia *vs.* adult NSC differentially expressed genes. We generated these gene sets by comparing the Borrett et al. E14.5 *vs.* P20-61 NSC cluster data. We first merged the differentially expressed genes from *Prdm16* cKO NSC *vs.* control NSC and radial glia *vs.* adult NSCs. Next, we made a radial glia-enriched gene set and an adult NSC-enriched gene set from the merged radial glia vs. adult NSC differentially expressed genes. We ran GSEAPreranked with the ranked *Prdm16* cKO NSC *vs.* control NSC differentially expressed genes to assess enrichment of genes from the radial glia-enriched gene set and adult NSC-enriched gene set.

### EdU staining

We used the Click-iT® EdU Imaging Kit (Invitrogen, C10337) for Edu staining. Mice received intraperitoneal injections of 50 mg/kg EdU at P14 and were sacrificed at P28. The brain sections underwent immunohistochemistry as previously described and were then incubated for 30 min with the Click-iT® reaction cocktail, prepared according to the manufacturer's protocol. After three washes with PBS, the brain sections were mounted and imaged using a Zeiss Apotome fluorescence microscope. Image analysis was performed using ImageJ.

### RNAscope in situ hybridization

P30 mice were transcardially perfused, ensuring RNAse-free conditions throughout the process. The brains were sectioned into 20 μm thickness after fixation with 4% PFA overnight, dehydration in 30% sucrose, and embedding in Tissue-Tek® O.C.T. compound. *Vcam1* signals were visualized following the RNAscope™ Multiplex Fluorescent Reagent Kit v2 according to the user manual (ACDBio, 323100). The Vcam1 probe (Cat No. 438641) was from ACDBio. The sections were imaged using a Zeiss Apotome fluorescence microscope and image analysis was performed using ImageJ.

### Quantification and statistical analysis

Image analyses were performed with the experimenter blinded to the genotype of the mice with the following exception: due to the enlarged ventricle phenotype of the *Prdm16* cKO mice, blinding of images including the ventricles was not possible. The numbers of mice and replicates are described in the figure legends. scRNA-seq data were analyzed as described above. All non-RNA-seq data analyses were conducted using R and the Prism 8 software (Graphpad). The normality of data was tested by the Shapiro–Wilk test. For data with normal distribution, Welch's $t$-test was used for two-group comparisons and one-way ANOVA with Tukey's multiple comparison test was used for multi-group comparisons. For data that deviate from a normal distribution, the Wilcoxon signed-rank test (for paired data) was used. Data from technical replicates from the same mouse were averaged and used as a single biological replicate in statistical tests. An estimate of variation in each group is indicated by the standard error of the mean (S.E.M.). * $p < 0.05$, ** $p < 0.01$, *** $p < 0.001$. Illustrations were generated with BioRender.

### Reporting summary

Further information on research design is available in the Nature Portfolio Reporting Summary linked to this article.

## Data availability

We deposited all RNA-seq data to the Gene Expression Omnibus under accession number GSE218159. Source data are provided with this paper.

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

## Acknowledgements

We thank Dr. Patrick Seale for sharing the PRDM16 antibody, Drs. Bruce Spiegelman and Paul Cohen for sharing the *Prdm16* flox mice, Drs. Fiona Doetsch, Daniel Levy, Lindsay De Biase, Michael Sofroniew, Harley Kornblum, and Aparna Bhaduri for advice, Amy Gleichman and Patrick Chen for technical assistance, Ranmal Samarasinghe for antibodies, Corey Harwell and Manuel Baisabal for reagents, and Brenda Urias for assistance with the illustrations. This work is supported by the NIH/NIMH T32MH073526 and T32 NS048004 and the Achievement Rewards for College Scientists Foundation Los Angeles Founder Chapter to M.I.G., the UCLA Eli and Edythe Broad Center of Regenerative Medicine and Stem Cell Research (BSCRC) post-doctoral training grant to J.L., the NIH/NINDS NS111378, NS117148 and the NIH/NICHD HD100298 to X.Y., the NIH/NINDS R00NS089780, R01NS109025, the NIH/NIA R03AG065772, NIH/NICHD P50HD103557, National Center for Advancing Translational Science UCLA CTSI Grant UL1TR001881, the BSCRC Innovation Award, the Friends of the Semel Institute for Neuroscience & Human Behavior Friends Scholar Award, the W.M. Keck Foundation Junior Faculty Award, UCLA Jonsson Comprehensive Cancer Center and BSCRC Ablon Scholars Award, Rose Hills Foundation Stem Cell Innovation Award to Y.Z.

## Author contributions

J.L., M.I.G., Y.L., and Y.Z. conceived the project, performed the experiments, and analyzed the data. A.J.Z. managed the mouse colony and performed some immunohistochemistry experiments. G.D., I.S.A., and X.Y. contributed to scRNA-seq experiments and data analysis. E.R., M.T., and S.T.C. contributed to experiments. B.G.N. contributed to the experimental design and provided reagents. A.C.-S. and A.A.B. assisted with the whole-mount experiments. J.L., M.I.G., Y.L., and Y.Z. wrote the paper. All authors read the manuscript.

## Competing interests

Y.Z. consulted for Ono Pharmaceuticals. All other authors declare no competing interests.
