## [Transparent Peer Review file · Nature Communications]

PRDM16 regulates the postnatal fate of embryonic radial glia via VCAM1-dependent mechanisms

Corresponding Author: Dr Ye Zhang

Version 0:

Reviewer comments:

Reviewer #1

(Remarks to the Author)

This manuscript is greatly improved especially by examining the phenotype of the PRDM16cKOs at later stages, up to 7 months, and adding the molecular analysis of direct PrDM16 targets regulated in the cKO.

It now makes a strong case for a novel role of PRDM16 in inhibiting NSC generation in lateral wall of the lateral ventricle, while neurogenic RGCs seem not to be prolonged in the cerebral cortex (Suppl. Figure 4). However:

- 1) It is therefore not clear why DCX+ transient neuroblasts continue to be produced. Maybe the authors could speculate about this in the discussion? In either case they should change the title, as the Dcx+ neuroblasts are transient and not giving rise to neurons in the cortex, and hence this work rather shows a "Role of PRDM16 in regulating adult NSC numbers in the lateral wall SEZ in a VCAM1-dependent manner", but cannot claim to prolong neurogenesis in other regions.
- 2) Moreover, as far as I could see for the cortex region no data from beyond P21 have been added, but claims are made about persistence of RGCs in adulthood. However, this is shown only for the neurogenic stem cell niche, the V-SVZ in the lateral wall of the lateral ventricle. Therefore either show stainings at later stages or tune down the statements about regions other than the known stem cell niche.
- 3) It is also not clear, how VCAM1 is regulated by PRDM16 (as it seems to be posttranscriptional). In this regard the authors should also rephrase lines 395/396 where they claim that PRDM16 regulates VCAM1. This occurs obviously only indirectly which needs to be stated and possibly the authors could propose an indirect mechanism from the candidates emerging from their RNA-seq data?
- 4) I am not convinced by the conclusions drawn from Figure 4D. The peak of the RGC-enriched genes is in the center – neither at the PRDM16 nor the WT side – so why do the authors claim these genes would be enriched in the PRDM16 KO cells?
- 5) The BrdU-label-retaining data shown in Figure 3 are nice, but it would be important to do a double-staining with VCAM1 to ensure that these label retaining cells are not ependymal cells, but rather stem cells.
- 6) Please also add information about the stages and regions analyzed to all Figure legends. For example, we don't know the age of the whole mounts shown in Figure 2, no information in the Figure 4 legend about when the RNA-seq analysis was performed (I could find the age of 1 months mentioned in the main manuscript, but please also add to the legend) and likewise in Supplementary Figures.

Reviewer #2

(Remarks to the Author)

The authors have performed extensive new experiments. Overall, the findings described in this manuscript are novel and significant. I therefore recommend the publication of this manuscript after minor revisions. Here are a couple of suggestions to further improve this manuscript: 1. Although the authors listed molecular pathways differentially expressed in each cell type in Prdm16 knockout mice, these results were not discussed in the context of the literature. For example, regulating cholesterol synthesis genes by prdm16 is novel in the context of emerging interest in the involvement of cholesterol in neurological disorders. Discussing how these findings advance our understanding beyond what is already known will improve this manuscript. 2. Supplementary figures 2 and 16 are partially redundant and should be revised.

Reviewer #3

(Remarks to the Author)

The revised manuscript is significantly improved and provides important insight into the function of Prdm16 in cortical progenitors. I support the publication.

Version 1:

Reviewer comments:

Reviewer #2

(Remarks to the Author)

The revised manuscript is significantly improved and provides important insight into the function of Prdm16 in cortical progenitors. I have no further question.

Point-by-point response to review

Reviewer #1 (Remarks to the Author):

This manuscript is greatly improved especially by examining the phenotype of the PRDM16cKOs at later stages, up to 7 months, and adding the molecular analysis of direct PrDM16 targets regulated in the cKO.

It now makes a strong case for a novel role of PRDM16 in inhibiting NSC generation in lateral wall of the lateral ventricle, while neurogenic RGCs seem not to be prolonged in the cerebral cortex (Suppl. Figure 4). However:

1) It is therefore not clear why DCX+ transient neuroblasts continue to be produced. Maybe the authors could speculate about this in the discussion?

We thank the reviewer for pointing out this important unanswered question. We added possible explanations of the phenotypes in the discussion as requested (page 20).

In either case they should change the title, as the Dcx+ neuroblasts are transient and not giving rise to neurons in the cortex, and hence this work rather shows a “Role of PRDM16 in regulating adult NSC numbers in the lateral wall SEZ in a VCAM1-dependent manner”, but cannot claim to prolong neurogenesis in other regions.

We thank the reviewer for helping us revising the title to more accurately describe our findings. We changed the title to “PRDM16 regulates the postnatal fate of embryonic radial glia via VCAM1-dependent mechanisms”.

2) Moreover, as far as I could see for the cortex region no data from beyond P21 have been added, but claims are made about persistence of RGCs in adulthood. However, this is shown only for the neurogenic stem cell niche, the V-SVZ in the lateral wall of the lateral ventricle. Therefore either show stainings at later stages or tune down the statements about regions other than the known stem cell niche.

We added “V-SVZ” to the description of the adult phenotypes throughout the manuscript to be accurate as requested by the reviewer.

3) It is also not clear, how VCAM1 is regulated by PRDM16 (as it seems to be posttranscriptional). In this regard the authors should also rephrase lines 395/396 where they claim that PRDM16 regulates VCAM1.

As requested by the reviewer, we changed lines 395/396 to “VCAM1 levels, post-transcriptionally regulated by PRDM16, thus determine whether radial glia stay in the embryonic radial glia stage or progress to the adult NSC stage...” We also added “indirectly” to the second sentence in the same paragraph, which now states “PRDM16 indirectly induces a reduction in VCAM1 level...”

This occurs obviously only indirectly which needs to be stated and possibly the authors could propose an indirect mechanism from the candidates emerging from their RNA-seq data?

Based on scRNAseq and ChIPseq analyses, we proposed an indirect mechanism through EGFR in the Discussion section (page 21).

4) I am not convinced by the conclusions drawn from Figure 4D. The peak of the RGC-enriched genes is in the center – neither at the PRDM16 nor the WT side – so why do the authors claim these genes would be enriched in the PRDM16 KO cells?

We apologize for not clearly specifying what was plotted in Figure 4D, thus causing confusion. We added clarification in the figure legend and included references that detail the methodology. Figure 4D legend now reads “PRDM16 cKO NSCs upregulated the expression of embryonic radial glia-enriched genes and downregulated the expression of adult NSCs-enriched genes. NES: normalized enrichment score. In GSEA, the enrichment score is calculated by “walking down” a ranked list of

genes, adding a value to a running sum when a gene belongs to the gene set being tested, and subtracting a value when it does not; the final enrichment score is the maximum deviation from zero encountered during this process, essentially reflecting how concentrated the gene from the gene set are at either the top or bottom of the ranked list^{70,71}. In the left panel, a positive enrichment score indicates that radial glia genes are enriched in the gene set containing genes upregulated in Prdm16 cKO mice. In the right panel, a negative enrichment score indicates that adult NSC genes are enriched in the gene set containing genes downregulated in Prdm16 cKO mice. The black bars in the middle panel represent individual genes. In the left panel, concentration of black bars on the left indicates that radial glia genes are enriched in the gene set containing genes upregulated in Prdm16 cKO mice. In the right panel, concentration of black bars on the right indicates that adult NSC genes are enriched in the gene set containing genes downregulated in Prdm16 cKO mice.

5) The BrdU-label-retaining data shown in Figure 3 are nice, but it would be important to do a double-staining with VCAM1 to ensure that these label retaining cells are not ependymal cells, but rather stem cells.

We thank the reviewer for an excellent suggestion. We co-stained BrdU with the NSC markers VCAM1 and GFAP and observed that most of the BrdU signal colocalizes with GFAP (see new Supplementary Figure 5). Because VCAM1 labels the apical surface of NSCs and BrdU labels cell nuclei, it was difficult to definitively determine whether adjacent VCAM1 and BrdU signals arise from the same cell in the V-SVZ, where cells are densely packed. Therefore, we have included only the BrdU/GFAP co-staining in the new figure.

6) Please also add information about the stages and regions analyzed to all Figure legends. For example, we don't know the age of the whole mounts shown in Figure 2, no information in the Figure 4 legend about when the RNA-seq analysis was performed (I could find the age of 1 months mentioned in the main manuscript, but please also add to the legend) and likewise in Supplementary Figures.

Added.

Reviewer #2 (Remarks to the Author):

The authors have performed extensive new experiments. Overall, the findings described in this manuscript are novel and significant. I therefore recommend the publication of this manuscript after minor revisions. Here are a couple of suggestions to further improve this manuscript: 1. Although the authors listed molecular pathways differentially expressed in each cell type in Prdm16 knockout mice, these results were not discussed in the context of the literature. For example, regulating cholesterol synthesis genes by prdm16 is novel in the context of emerging interest in the involvement of cholesterol in neurological disorders. Discussing how these findings advance our understanding beyond what is already known will improve this manuscript.

We thank the reviewer for their constructive criticism. We added a paragraph in the Discussion section on the novel insight from our scRNAseq analyses (page 21-22).

2. Supplementary figures 2 and 16 are partially redundant and should be revised.

We have combined these two partially redundant figures.

Reviewer #3 (Remarks to the Author):

The revised manuscript is significantly improved and provides important insight into the function of Prdm16 in cortical progenitors. I support the publication.

Point-by-point response to review

Reviewer #2 (Remarks to the Author):

The revised manuscript is significantly improved and provides important insight into the function of Prdm16 in cortical progenitors. I have no further question.

Thank you.